# Insights and inspirations: A qualitative exploration of community health workers' motivations in Myanmar and Bangladesh

**Nyo Yamonn** *, **Catherine Lee**, **Tom W. J. Y. Traill**

Community Partners International, Bangkok, Thailand

* nyoyamonn@cpintl.org

## Abstract

Community Health Workers (CHWs) play significant roles in various settings, with their motivations and retention strategies widely studied. Yet, literature is sparse on CHWs from Myanmar, who are key to primary health care in marginalized and conflict-affected areas. This study explores the unique challenges these CHWs face, using firsthand accounts. Life story interviews, enhanced with a lifeline tool, were conducted with 34 CHWs from conflict-affected regions in Myanmar and in Rohingya camps in Bangladesh. Additionally, eight key informant interviews were held with leaders from organizations that work with CHWs. Data analysis was facilitated by NVivo 14 software and four layers of influence adapted from Urie Bronfenbrenner's ecological systems theory of human development. The findings reveal that, CHWs primarily joined organizations to acquire skills and knowledge. In Bangladesh, the focus was on job-related skills, whereas in Myanmar, healthcare skills were prioritized. Despite remuneration being inadequate, it remained crucial for retention, as did the sense of being valued by the community in Myanmar. Mental health support emerged as a potential need for CHWs. Funding deficits and fragmented support presented organizational challenges, thereby impacting both program implementation and retention of CHWs. To address these challenges, effective, sustainable CHW programs in conflict-affected regions require a shift towards long-term support for organizations and health systems. This includes focusing on CHWs' mental health and stakeholder engagement. Short-term, fragmented solutions may revert to pre-existing situations once removed. Sustainability planning is key to break the CHW turnover cycle and maximize investments in these contexts.

## Introduction

In 2021, Myanmar faced abrupt political changes when the military overthrew the elected government. Protests erupted all over the country, with many government staff participating in the Civil Disobedience Movements (CDM) [1]. The system of law and order was disrupted, and all civilians were at high security risk during the period of political turmoil. Due to arrests, torture and interrogations, individuals who participated in political movements fled the

**Data Availability Statement:** The data cannot be made publicly available due to its sensitivity. The datasets used and analyzed during the current study are available from the Chief Executive

Director of Community Partners International, Dr. Si Thura, at sithura@cpintl.org, upon reasonable request.

**Funding:** The study is funded by Community Partners International's Core Funds. The funders were not involved in any part of the study or the preparation of the manuscript.

**Competing interests:** The authors have declared that no competing interests exist.

country. Many of them sought refuge in various areas controlled by armed groups and neighboring countries [2].

Even before the coup, conflicts of varying intensity and frequency between ethnic armed organizations (EAOs) and the military, as well as among EAOs, existed in Myanmar for over seven decades [3]. Despite a relatively politically stable period from 2012 to 2020 brought about by ceasefire negotiations and agreements, bouts of clashes and conflicts between the Myanmar Military and armed groups still occurred [4]. After the 2021 political changes, all existing conflicts increased in terms of intensity and frequency, and the affected areas also expanded [5].

For several decades, Ethnic Health Organizations (EHOs) and Community-Based Health Organizations (CBHOs) in conflict-affected areas of the country have maintained their own health systems to address the needs of their communities. They have trained and deployed community health workers (CHWs) who play crucial roles in providing health services in conflict-affected and hard-to-reach areas [6,7].

According to the World Health Organization, CHWs are healthcare providers living in the communities they serve. They typically have less formal education and training compared to professional healthcare workers such as nurses and doctors [8].

Globally, CHWs are recognized for their significant potential to extend healthcare services to underserved populations, including remote communities and historically marginalized groups. They play a vital role in addressing unmet health needs in a culturally appropriate manner, enhancing service accessibility, reducing health disparities, and improving the overall performance and efficiency of the healthcare system [9].

CHWs from EHOs in Myanmar operate with limited resources, both in terms of personnel and materials. Therefore, during relatively stable political periods, EHOs in Myanmar initiated discussions for health system convergence with the Ministry of Health of the Myanmar government and established referral networks with government hospitals and clinics [10]. However, after the 2021 coup, the participation of many government staff in CDM caused many of these referral networks to cease functioning.

Additionally, the increasing number of internally displaced people in their areas put further pressure on the CHWs to not only maintain existing services but also expand coverage to address the health needs of those newly displaced. Consequently, we aimed to explore how they are coping and managing this situation with their existing resources in the amidst of conflicts and political unrest. Moreover, previous studies have not delved into their life stories and listened to their voices at a deeper level.

The primary objective of our overall study is to investigate the support mechanisms for CHWs from EHOs in Myanmar and community-based health programs in Rohingya, Bangladesh. However, in this article, we focus only on the motivations and retention of CHWs.

Leveraging our existing connections and capabilities, we selected two conflict-affected areas of Myanmar for our study: Kayin State and Southern Shan State. We also included the Rohingya camps in Cox's Bazar, Bangladesh.

Although the Rohingya camps in Cox's Bazar operate in a relatively stable environment compared to the conflict-affected areas of Myanmar, they occasionally face gun violence, arson, and extortion due to the presence of armed groups in their areas [11]. Despite currently residing in Bangladesh, all the CHWs and the communities they serve originally came from Myanmar and, thus, are relevant to include in our study.

While our focus is on these specific regions, our ultimate aim is to illuminate the situation for CHWs in conflict-affected areas more broadly. We posit that the life experiences of CHWs are a valuable data source that can inform strategies to enhance their support systems, not only in Myanmar and the Rohingya but also in other conflict-impacted regions.

## Study context

Our study was conducted in Kayin State and southern Shan State of Myanmar, as well as Cox's Bazar camps in Bangladesh.

Myanmar, located in Southeast Asia, is home to more than 135 diverse ethnic groups. Approximately one-third of the townships in Myanmar were already home to various armed groups before the military coup in 2021 [10]. These areas experienced periods of armed clashes and political unrest throughout history, predating Myanmar's independence. Moreover, many of these regions pose challenges for access due to both geographical and political obstacles, making the communities heavily reliant on healthcare services provided by EHOs and CBHOs [6,7]. Health indicators have already demonstrated the impact of conflicts in areas of Myanmar where EHOs and CBHOs operate. For instance, in these areas, the under-five mortality rate is 141.9 per 1000 live births, compared to the national average of 46.1 per 1000 live births in Myanmar, according to the latest available data from 2011 [12,13].

Given the scarcity of healthcare professionals and the challenges for ethnic minorities in the government health system, CHWs from these organizations have taken on the responsibility of delivering primary healthcare to these communities [12]. In contrast, in Cox's Bazar, Bangladesh, Rohingya CHWs are limited to health education, community mobilization, and referral roles in their assigned camp areas. They can only play limited roles due to the rules imposed by the Bangladesh government [14].

Prior to the coup, approximately 118 of Myanmar's 330 townships, containing more than 12.4 million people, which is 23% of total population, were affected by conflict and tensions between EAOs and the government [3]. After the coup, there was renewed fighting between EAOs and the government's armed forces and more people were displaced. In addition, there were conflicts among EAOs, particularly in Shan State [5].

Among the conflict-affected areas, the eastern part of Myanmar is well-known to be one of the most impacted [12]. Our study areas in Myanmar are situated within the regions of eastern Myanmar where various armed groups operate.

Kayin State has one of the strongest insurgency groups and the longest history of conflicts with the central government while striving for greater autonomy and recognition of their rights. Although initially united, the insurgency group has experienced several splits over the years. As a result, Kayin Sate now has more than one armed group, all of which originated from the same group [15].

Shan State has a rich history of separate sovereignty under its own kingdoms and diverse ethnic groups. The region was once ruled by autonomous local chiefs known as Saophas. During British colonial rule, Shan State became part of British Burma, retaining some local autonomy. After Myanmar gained independence in 1948, Shan State joined the Union of Burma, and the Saophas were gradually removed from power. Since then, the state has experienced armed conflicts between ethnic armed groups seeking autonomy and the central government [16–18].

In Bangladesh, approximately one million Rohingya refugees from Myanmar reside in Cox's Bazar, the world's largest refugee settlement. In August 2017, more than 700,000 refugees from Myanmar sought refuge in Cox's Bazar to escape the intense violence and conflicts they had been enduring. The remaining refugees arrived prior to this timeframe [2]. UN Secretary-General Antonio Guterres stated in a press conference that the Rohingya people are among the most discriminated against globally [19]. At present, Rohingya refugees in Bangladesh have established networks within the camps, involving volunteer educators, healthcare workers, and aid distributors [20]. Community health workers, who are known there as community health volunteers, play a vital role in providing support. They serve as a significant

solution to bridge the gaps that arise from language, cultural, and belief barriers within the Rohingya community [21].

## Methodology

To explore the life and work experiences of CHWs included in this study, our research employed life story interviews together with lifeline visualization. The life story is a qualitative research technique used to collect comprehensive information about the fundamental aspects of a person's entire life. This method enables an in-depth examination of individual life experiences [22]. Lifelines serve as visual aids that illustrate the progression of life events in a time-ordered manner, and they may encompass assessments of these occurrences [23]. Narrative research methods, such as life story interviews and lifeline visualization, offer deep insights into research subjects and are valuable in understanding interactions and ethical responsibilities [24]. Despite their long history in anthropology and sociology, they are relatively new and underutilized in health research [25,26]. Utilizing visual elicitation techniques in semi-structured or in-depth interviews addresses various challenges such as rapport building, managing power dynamics, and participant considerations. These methods empower research participants by positioning them as experts, thereby enhancing trust and engagement while mitigating power imbalances [27]. These visual methods enhance data quality and provide fresh insights, especially for diverse populations facing complex situations [28–30]. Incorporating visual media, enriches dialogue and aids data interpretation, fostering validation and empowerment through a participant-centric approach [31]. Lifelines help interviewees express themselves effectively, even on challenging topics by providing emotional distance from traumatic events [30]. Furthermore, combining lifelines with life story interviews enriches data and promotes critical thinking [32,33].

## Materials and methods

### Ethics statement

This study received ethical approval from the Community Ethics Advisory Board (CEAB) based in Mae Sot, Thailand. The board is comprised of representatives from the EHOs and CBHOs, along with technical experts. All participants provided informed consent, and their confidentiality was maintained throughout the study.

### Study tools

The primary focus of the study was on life story interviews (LSIs), which incorporated lifeline tools. Key Informant Interviews (KIIs) were also conducted to provide an overview of the situations faced by the CHWs with whom they work.

For LSIs, we used an open and semi-structured question guide to compose the life stories. LSIs began with background information and the lifeline drawing process. Visual references, including sample lifelines with a horizontal line separating positive and negative life events, were provided to assist participants in mapping their life journey from birth. We prioritized visual representation over polished graphs. KIIs were conducted by using a semi-structured questionnaire.

### Study participants

For LSI, CHWs, irrespective of their specific titles, including both trainees and active workers aged 18 years and older, were selected in each of the three study areas using purposive sampling. LSI Participants were deliberately selected with a focus on diversity in terms of

geographical areas, organizations, sex, work experiences, types or levels of health workers/volunteers, and years of experience to have the maximum variation.

In the early stages of the study, focal points at each organization were provided with specific criteria regarding the characteristics of the CHWs we aimed to interview, including their type and level of work, years of experience, gender, and assigned geographic area. These focal points facilitated the process by initially contacting potential participants. As the study progressed, data collectors secured authorization from most organizations to communicate directly with potential participants. Consequently, the data collectors approached individuals from these organizations directly to request their participation in interviews.

KII participants were selected using convenience sampling and included those in leadership positions of organizations working with or training CHWs. KII participants were identified by the study team and contacted directly by the principal investigator (PI).

## Study team

The study team included the PI, a native of Myanmar with strong local knowledge, community health expertise, and qualitative research experience. The data collectors, proficient in local languages and experienced in qualitative data collection, were familiar with their respective study areas in Myanmar and Bangladesh. Additionally, experienced researchers provided valuable input on research design, methodology, and data analysis. The team's combined expertise in the local context and qualitative data collection ensured effective and culturally sensitive data gathering.

To address potential psychological trauma among CHWs, data collectors underwent comprehensive training. This training covered psychosocial considerations, research ethics, data collection procedures, tool usage, and mental health support resources.

Individual training sessions, conducted online by the PI due to their geographic dispersion, lasted approximately two days each. Following the training, data collectors received detailed guidance and had regular individual meetings throughout the process. The study team also established contact with mental health support providers before data collection, ensuring immediate access for referrals if necessary.

## Consent process

Participants were provided with a comprehensive explanation of the study's purpose and procedures in accordance with CEAB-approved informed consent guidelines. They consented either through signed forms or recorded verbal agreement, depending on the research context.

## Study design

The study commenced with KIIs, and all interviews were conducted online by the PI and the data collector from Bangladesh.

Initially, our plan was to conduct all LSIs online. However, discussions with participating organizations revealed significant challenges in conducting online interviews in Myanmar, including limited phone and internet networks, and security concerns. To ensure safety and address these challenges, we conducted all LSIs except three in person. In Bangladesh, after piloting the lifeline tool online, we discovered a preference for in-person interviews. As a result, all LSIs in Bangladesh were conducted face-to-face.

The LSIs were conducted using a semi-structured questionnaire. Participants were initially asked background questions before creating lifeline drawings. For these visualizations, participants used A4 paper and pens or pencils, and had approximately 30 minutes to complete their drawings independently, with sample drawings provided for reference. These visualizations

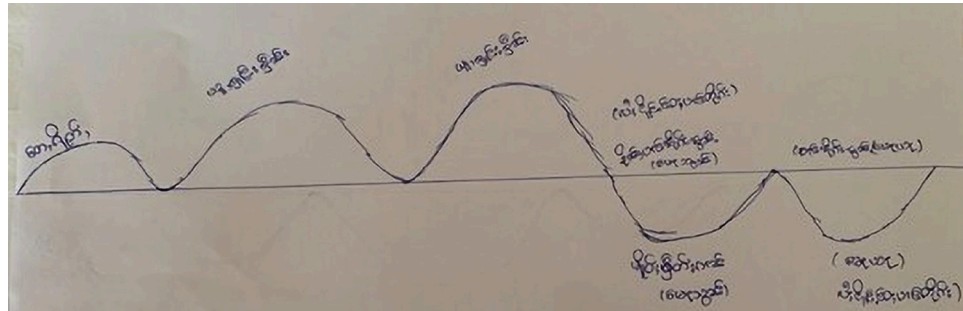

**Fig 1. A lifeline drawn by a participant from Shan State.**

were done in private and familiar settings, such as participants' offices or designated community spaces.

The lifeline interviews were conducted by trained local data collectors who were fluent in the local languages and familiar with the participants' cultural context. The interviews took place in similar private and comfortable settings. During these interviews, participants explained the significance of their lifeline drawings, and the data collectors used probing questions to explore their stories in depth. The data collectors were guided on maintaining self-awareness and minimizing biases, and regular debriefing sessions were held to discuss their experiences and refine the interview process.

The lifeline tool effectively facilitated meaningful conversations, helping community health workers recall and reflect on significant life events. Although adherence to instructions and levels of participation varied, most participants completed their drawings independently without requesting additional support.

The questions were translated into participants' languages by the data collectors, but full written translations were not provided except Myanmar Language due to the absence of a standardized written form of the Rohingya language and the limited proficiency of the data collectors in written Kayin and Shan languages (Fig 1).

Each data collector conducted and transcribed the first interview for review by the PI to ensure questionnaire compliance. Subsequent interviews were conducted in batches without immediate transcription and submission, considering time constraints.

After conducting approximately half of their interviews, interviewers were instructed to temporarily pause data collection and prioritize transcription. This approach ensured alignment with the study's intended goals and allowed for adjustments to questions, the data collection and ethical process if necessary.

## Data saturation

In our study, we selected 34 participants from three different geographical areas for a comparative analysis. To ensure a diverse range of voices from each area, we selected both male and female participants with varying levels of experience, roles, and assigned areas. As a result, our sample is not homogeneous in nature. Although we achieved thematic saturation for the major themes, a few new themes continue to emerge. However, due to safety and security concerns, we had to cease our data collection.

## Analysis

Transcripts from both the Kayin and southern Shan study areas were translated into the Myanmar language, while transcripts from the Bangladesh study area were translated into

English. All transcripts were de-identified and then thematically coded using NVivo14 software.

Due to limited human resources, we adopted a single-coding approach, carried out by the PI, who was chosen for her availability, bilingual abilities, and deep understanding of the research context. The initial coding, guided by the research objectives, revealed that some codes were too broad or missed important nuances, leading to a second round of refinement. This iterative process allowed for more accurate capture of emerging themes. The same researcher coded and recoded the data three times, reconciling differences by revisiting data segments and refining the coding scheme. This approach ensured the final coding reflected the bilingual nature of the data and enhanced the reliability of our analysis for the write-up.

To analyze the factors influencing the motivations and retention of CHWs, we employed four layers of influence adapted from Urie Bronfenbrenner's ecological systems theory of human development, simplifying it to personal, community, organizational, and external layers [23] (Fig 2).

## Results

### Participants and completed interviews

Data collection occurred from May to November 2022, with life story interviews conducted from June to November 2022.

The study comprised 34 LSIs conducted alongside lifeline tools with CHWs from conflict-affected regions in southern Shan State and Kayin State (hereafter referred to as Shan and Kayin, respectively) of Myanmar, as well as those in Cox's Bazar camps in Bangladesh for comparative analysis. Respondents participated from two organizations in Kayin, three organizations in Shan, and two camps in Bangladesh (Table 1). Additionally, 8 KIIs were conducted with individuals in leadership positions within organizations that work with or provide training to CHWs.

Below are three insightful stories, one from each region, selected to shed light on the varied challenges faced by community health workers. These accounts, shared under pseudonyms to protect confidentiality, may not capture all experiences but offer a meaningful glimpse into their respective contexts.

**Khadija,** a 24-year-old, worked as a Sexual and Reproductive Health Volunteer in the Rohingya camps of Cox's Bazar, Bangladesh. Her journey began as a Rohingya escaping intense conflicts, walking from Rakhine to Cox's Bazar in 2017. She vividly shared enduring four days of starvation before reaching safety in Bangladesh.

After five months in Bangladesh, Khadija secured a job with an NGO, marking a turning point in her life. She recalled the experience of when she and her sister handed their first salaries to their father as a moment filled with emotion.

*"We handed our first salaries to our father, saying, 'Father, this is our first salary.' My father hugged us and cried loudly. I cried too. For 14–15 years, my father earned, and we always took money from him. But today, I fulfilled my promise to my father. I cried a lot. I don't know if they were tears of happiness or sorrow, but I just shed tears. All the members of the family cried that day. From that day, I knew that women can be strong and leaders on their own."*

Securing employment was a privilege, especially considering she never anticipated being able to financially support her parents during a family crisis.

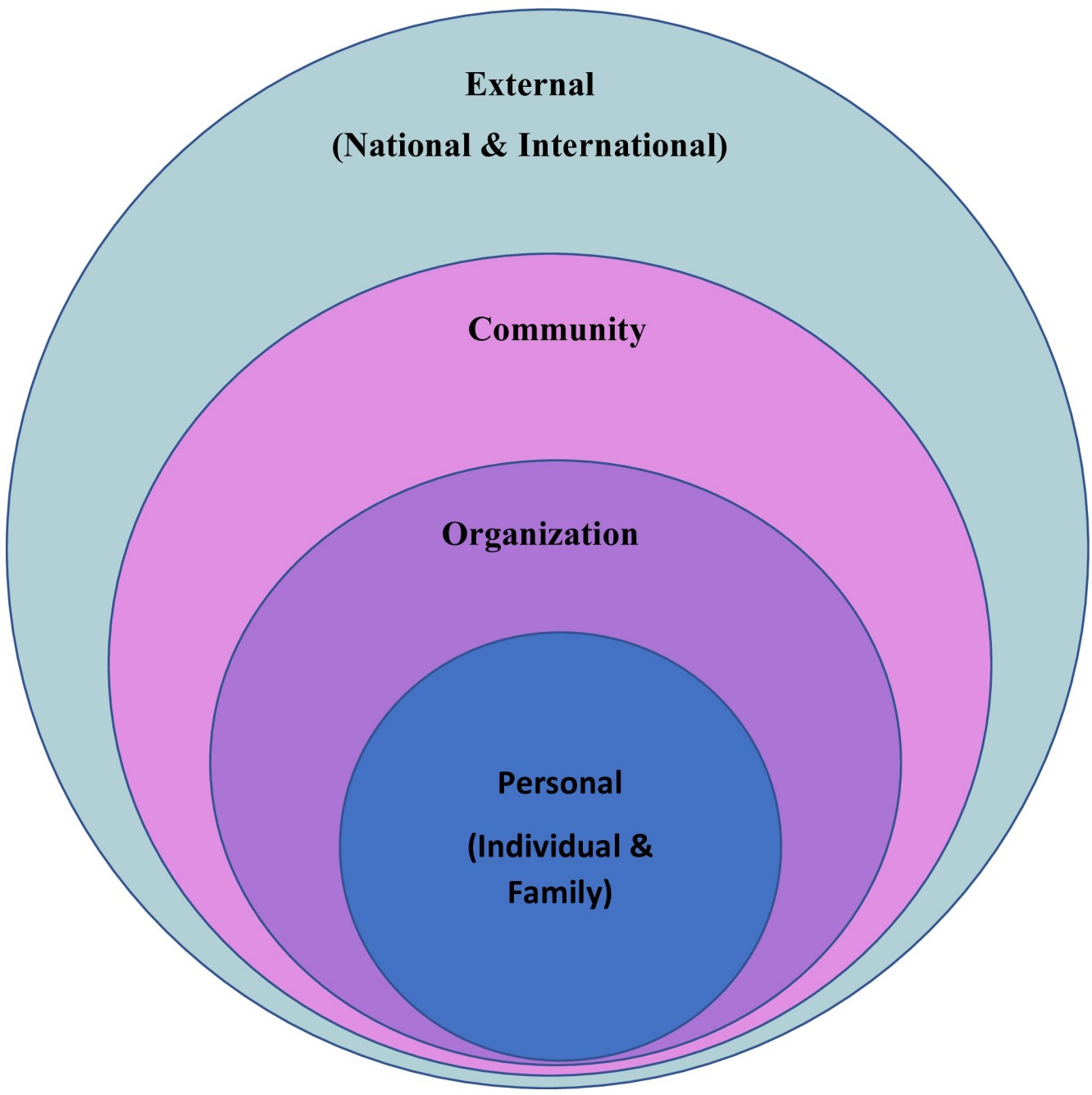

**Fig 2. Four layers of influence.**

Due to personal security concerns, Khadija made the difficult decision to marry but continued her work, challenging societal expectations. Fortunately, her husband supported her aspirations, allowing her to persist in her career.

Khadija actively conducted awareness sessions on reproductive health and family planning. Additionally, she provided home visits to pregnant mothers, ensuring they deliver in a health facility. Despite initial resistance from some community members, she earned respect and witnessed positive changes in attitudes over time.

**Table 1. Participants' characteristics.**

| Geographical Areas | Organization/Camp | Work Experience | Sex (F = Female, M = Male) |
|---|---|---|---|
| **Kayin** | 2 Organizations: 12 Participants | <1yr | 2 F |
| | | 1-5yrs | 3 F, 3 M |
| | | >5yrs | 2 F, 2 M |
| **Southern Shan** | 3 Organizations: 12 Participants | <1yr | 1 F, 1 M |
| | | 1-5yrs | 3 F |
| | | >5yrs | 4 F, 1 M |
| **Bangladesh** | 2 Camps from Organization: 10 Participants | <1yr | 2 F, 2 M |
| | | 1-5yrs | 2 F, 4 M |
| **Total** | **6 Organizations: 34 Participants** | 8 = <1yr 15 = 1-5yrs 11 = >5yrs | 19 F, 15 M |

Khadija's commitment to her job was fueled by transformative training and the supportive environment she experienced. Despite facing challenges, she appreciated the justice and integrity within her organization.

**Naw Say,** a 26-year-old Medic Trainee from Kayin, comes from a farming family of nine. Limited educational opportunities in her village prompted her enrollment in a charity school in a distant city at the age of 10. Following high school, local policy obligated her to serve her community through an organization. Choosing to become a CHW, she initially assisted in the clinic before undergoing CHW training.

Assigned to a frontline village, Naw Say's clinic encountered airstrikes, leading to perpetual readiness and makeshift treatment areas in the forest. Drawing from past evacuations, she instinctively responded to airplane sounds even in safe situations. Inspired by CHWs visiting her village, she chose this path over other options due to its perceived ease. With over five years of service, her dedication was fueled by the encouragement of her supervisor. She said, *"Continuing my work is fueled by the consistent encouragement from my supervisor, who holds a significant role in guiding and supporting me. His words of encouragement matter greatly, and even in the absence of additional assistance, his support keeps me motivated to proceed with my work."*

Challenges for Naw Say included limited skills, strict organizational disciplines, and occasional criticism from some community members for taking on a role traditionally reserved for men. Despite these hurdles, she persevered, grounded in the broader purpose of her work. Living five hours away from her family, she couldn't visit them even during their illnesses due to work commitments. Irregular incentives further contribute to the hardships she faced.

Caring for patients, including those with serious trauma amid conflicts, Naw Say and her colleagues confronted resource shortages due to destroyed supplies. Prioritizing critical cases, they could only provide health education for minor illnesses. Naw Say advocated for additional training to address community needs effectively. Despite challenges, her hope persists to reconstruct the clinic and enhance patient care since its destruction.

**Nang Noon,** an 18-year-old Community Health Worker from Shan, was motivated to enter her challenging path by family encouragement and a desire to address the lack of available medical service providers. With inspiration from a cousin already in the organization, she joined and has served as a CHW for two years.

Nang Noon's commitment extended beyond her duties, driven by a desire to expand her knowledge and improve healthcare skills. Her motivation to specialize in maternal and child healthcare was fueled by the alarming trend of young women's mortality during childbirth.

Due to the considerable distance to other healthcare centers with difficult roads and financial constraints, they become the primary accessible healthcare provider and the sole option for the community, earning acceptance and respect even among the elders—a source of solace for Nang Noon.

Despite available referral fee support for pregnant mothers, Nang Noon often covered transportation fees for other cases from her own money due to limited alternatives.

Navigating through various modes of transportation and carrying heavy supplies, Nang Noon faced numerous challenges, especially during the rainy season. She experienced motorcycle accidents and even narrowly escaped a potential disaster of going into an abyss.

Workplace challenges compound her burdens, with criticism, public reprimands, and mistreatment heightening her exhaustion and sometimes bringing her to tears.

*"Once, I made a mistake without realizing it. I was loudly reprimanded in front of the patients, even though I was unaware of what had happened. I felt deeply embarrassed and disappointed at that moment."*

Her income proved inadequate even for personal needs, frequently requiring financial assistance from her family. To address staff turnover, she advocates for salaries that cover basic needs and mitigate the strain on healthcare workers.

The following analysis presents both LSI and KII data across personal, organizational, community, and external levels. Please see the summary of findings in the Table 2.

## Motivations to become a CHW

**Personal level.** *Income needs*. In Bangladesh, six out of ten CHWs mentioned that the need for income to support their family was a primary motivational factor for them to become CHWs. A CHW born in a camp in Cox's Bazar shared that she had passed interviews to attend a university program but had to forgo the opportunity to earn money to support her family.

**Table 2. Overview of themes across four layers of influence.**

| Four Layers of Influence | Themes |
|---|---|
| **Motivations to become CHW** | |
| Personal Level | Knowledge and Skills |
| | Income Need |
| | Want to Help the Community |
| | To Forget the Past |
| External Level | Compulsory Service |
| **Motivations to continue as a CHW** | |
| Personal Level | Compensation |
| | Additional Skills |
| | Lack of Better Options |
| Organizational Level | Compensation |
| | Work Environment, Supervision and Management |
| | Logistic and Referral Support |
| | Psychosocial Impact |
| Community Level | Feeling Needed by the Community |
| External Level | Safety and Security Issues |
| | Adequate and Timely Support from Donors and INGO |

Her mother has been ill since she was 12, and she has had no knowledge of or contact with her father from a very young age.

> *"I have interviewed at the Asian University for Women three times, and each time, I was successful. They were impressed and asked me to prioritize my life over my family. However, I cannot do that. How can I abandon those who are helpless? I may not succeed in doing so. Blessings play a crucial role in people's lives. Even if I were to go there, my thoughts would constantly drift towards my mother, wondering how she is coping. My concentration would be compromised. If we had someone to support us, I would have taken the opportunity to go there."* **22 years old female Sexual and Reproductive Health Volunteer, Bangladesh**

In Kayin and Shan, where incentives were irregular or not enough, income needs were rarely a motivation. Only one CHW from Shan mentioned it as a reason for joining his first CHW job with a different organization from a different region.

*Knowledge and skills.* CHWs in Kayin and Shan need to provide healthcare to the community, but in Bangladesh, the CHWs are restricted from doing so because of government restrictions, and their roles are limited to health education and referral. In Shan and Kayin, half of the CHWs chose this role due to their interest in medical care and acquiring health care skills, while in Bangladesh, half of the CHWs were motivated by the desire to gain job-related skills and health information.

*To help the community.* In Kayin, health care needs of the community were a significant concern for many individuals who joined the organizations. Half of them stated that one of the reasons for joining their organization was due to their understanding of the healthcare gaps in the community, and their desire to make a positive impact. Additionally, 11 out of 12 expressed that they joined their respective organizations with the intent to contribute to healthcare services for their community or their own family.

In addition, CHWs from both Kayin and Shan mentioned that difficulties accessing healthcare in the areas were also reasons for becoming CHWs, driven by a desire to care for themselves, their families, and their communities.

Two CHWs from Bangladesh expressed their desire to help their community as their primary reason for becoming health workers.

> *"When we were in Myanmar and experiencing torture, there was no one to assist us as everyone was rushing to flee. When I see someone in pain here, it reminds me of those old days, so I try to help people as much as possible, both physically and mentally."* **20 years old female Sexual and Reproductive Health Volunteer, Bangladesh**

*To forget the past.* In Bangladesh, a few CHWs mentioned that working in an organization can preoccupy their minds and help them avoid dwelling on their past distress. Except for one CHW who was born in the camp, all of them have experienced active conflict situations in Myanmar and endured days of walking to reach Bangladesh. One CHW revealed that they went without food for four days, while two others mentioned witnessing people being killed in front of them. Moreover, they have faced discrimination throughout their lives and missed out on further educational opportunities.

> *"I wanted to engage in something to keep myself from thinking about the trauma."* **39 years old male Community Immunization Volunteer, Bangladesh**

*"We got a job in the NGO which was a huge support for us. My mind became fresh again from the violence and suffering we faced after getting the job."* **24 years old female Sexual and Reproductive Health Volunteer, Bangladesh**

Despite facing psychological challenges, CHWs in Kayin and Shan did not cite these as motivations.

### Organizational level

**Recruitment process.**   Only the organization in Bangladesh used an open and formal recruitment process. One organization from Shan stated that they were working to establish an open and formal recruitment process whenever possible, but they also engaged in internal recruitment without a formal process. Other organizations from both Kayin and Shan relied on influential individuals from the community and collaborated with local ethnic and community-based organizations in the area for recruitment.

### External level

**Compulsory service.**   Two female CHWs from Kayin and two male CHWs from Shan became CHWs due to the compulsory requirement imposed by their local authorities to serve the community. These requirements vary over time and across different areas. They chose to become CHWs because it was considered a relatively better option than serving in other assigned roles.

### Motivations to continue as a CHW

**Personal level.**   *Compensation*. In Bangladesh, satisfaction with the current remunerations appeared to be major factors for staying in the organization. All participants except two said their incentives were enough to support their family. One expressed they understand the restriction of the amount of their incentives by the guidelines of the Bangladesh government. A few said it is enough even if someone gets sick, but one of them said,

*"It covers our expenses if we spend less, but it doesn't cover it if my children get sick or need treatment. We can't eat or wear as much as we want. We can't afford to eat fish or meat, as it would make it difficult to manage household expenses. We opt for leafy vegetable alternatives to fish and egg alternatives to meat."* 36 years old male Community Health Volunteer, Bangladesh

On the contrary, participants from both Kayin and Shan explained the insufficiency of incentives.

In Kayin, CHWs faced inconsistent salaries tied to donor projects, with some receiving no pay and others benefiting from pooled funds. These funds, in addition to covering salaries, were used for clinic supplies. Some CHWs contributed 10% of their project salary to support colleagues not covered by the project. Single CHWs relied on organization-provided meals or dried food supplies, while others, due to demanding work hours, struggled to find additional sustenance from external sources.

In Shan, nearly all CHWs voiced concerns about the insufficiency of their salaries, leading to financial challenges in supporting themselves and their families. To cope, many resorted to seeking financial assistance from family members, and a few engaged in additional income-generating activities such as agriculture and animal breeding. Some CHWs received supplementary dried rations alongside their salaries. Two of them said they wanted to leave their job

in the future to have better income. One other CHW explicitly stated, *"The help I need is an increase in my salary. I am 27 years old and getting older, and I should be able to provide income for my father, but currently, I have to ask for money from him. Additionally, I am responsible for providing for our needs such as rice, oil, and salt, and my income is not sufficient to cover these expenses. Furthermore, there is no one else in the family who can contribute to the income, and my two younger sisters are still in school."* **27 years old female Maternal and Child Health Worker, Shan**

*Additional skills.* Five CHWs in Shan expressed that they wanted to learn more about healthcare, aimed to reach the next level, or wished to improve their health-related skills. It appeared to be the main reason for staying in their jobs.

Half of the CHWs in Kayin also cited the desire for additional skills as a reason for staying in their jobs. Moreover, an organization leader from Kayin noted that one CHW left due to a lack of opportunities for advanced training.

In Bangladesh, CHWs expressed satisfaction with the skills development and training provided by their current organization, which motivated them to stay in their roles.

*Lack of better options.* In Shan State, CHWs stayed in their jobs due to a lack of better options, with seven considering leaving if opportunities arose. Similarly, some CHWs in Bangladesh, living as refugees with employment restrictions, stayed in their jobs due to limited alternatives. One CHW's attempt to open a shop was thwarted by government's regulations. No similar responses were given by respondents in Kayin.

**Organizational level.** *Compensation.* Leaders from Kayin and Shan organizations reported high turnover, except for one organization that offered a low but consistent salary, along with provisions such as dried food and accommodation for CHWs. This organization, despite these benefits, estimated a turnover rate of around 20% and noted that CHWs typically fulfilled their committed years of service before leaving. In Kayin, the introduction of support in the form of stipends and food by an organization resulted in longer retention, with CHWs staying for periods ranging from 4 to 6 years, compared to the previous range of 1 to 3 years. Meanwhile, the Bangladesh organization reported a very low turnover rate, with departures typically due to pursuits such as higher education, marriage, or relocation.

*Work Environment, Supervision and Management.* In Shan, the CHW featured as the story expressed a desire to leave her position due to her workplace situation. She faced day-to-day verbal abuse and mistreatment from her superiors. She mentioned she was not the only one facing that and there were CHWs who left because of that reason.

In Kayin, two CHWs highlight the importance of supportive supervisors in their decision to continue working.

*"I find motivation to stay in my job through a clear understanding of my superiors' intentions in providing continuous training and support. I am dedicated to my career in healthcare, acknowledging the purpose behind the training sent to me by my superiors."* **23 years old female Medic Trainee, Kayin**

One leader from an organization in Shan and both leaders from organizations in Kayin mentioned that one of the reasons for staff departure is related to management issues. The Shan organization discovered this trend through exit interviews, where individuals cited reasons such as a lack of respect in the workplace and dissatisfaction with organizational systems, including policies and practices. A Kayin organization estimated approximately 15% of health workers leave due to difficulties in establishing positive relationships with their superiors.

In Bangladesh, being content with the work environment and supervision appeared to be another factor contributing to job satisfaction. All participants expressed satisfaction with their work environment, maintaining positive relationships with colleagues and supervisors, and mostly receiving needed support.

*Logistic and Referral Supports*. Supply shortages in Kayin and Shan affected CHW work and morale.

Eight out of twelve CHWs in Kayin indicated that insufficient medical and material supplies significantly impacted their current or past work. One CHW highlighted the destruction of medicines, medical supplies, and materials caused by air strikes, resulting in loss of crucial resources. Another CHW expressed that the lack of organizational support in providing vehicles for transporting medical supplies on a regular basis acts as a barrier to effective work.

*"For supplies, it would be beneficial if we were supported by medicines. With such support, we can train individuals under our guidance, enabling them to work independently. Increasing the availability of supplies would allow us to expand our training efforts. However, in the absence of adequate supplies, even if we are training individuals, we will not be able to retain them."* **26 years old male Medic Trainee, Kayin**

*"Furthermore, it would be beneficial to receive additional support in terms of medical materials. This is crucial as our patient treatment extends beyond the confines of a hospital or clinic. At times, we are required to provide medical care to patients with traumas in forested areas. Unfortunately, in such situations, we are unable to clean medical materials immediately after use. These materials often need to be left unattended for 3 to 4 days, rendering them no longer suitable for reuse."* **38 years old male Senior Medic, Kayin**

Eight CHWs in Shan States mentioned that they do not have enough medicines to provide treatment to the community. Some noted that, while there is support for medicines and medical supplies, these resources do not meet their specific needs. Two of them stated that they regularly purchased medicines with their own money to treat patients. They also had to cover certain clinic expenses, such as fuel fees for the generator, from their salaries. Three CHWs mentioned that they needed to provide referral fees for patients from time to time. One CHW pointed out that they do not have sphygmomanometers. Consequently, they provide treatments without measuring blood pressure, which poses a danger to the patients.

No similar issues were reported in Bangladesh.

*Psychosocial Impact*. Both stress and tiredness were mentioned as factors for CHWs wanting to leave their jobs in the Shan study area. Both Kayin and Shan areas experience shortages of health workers, contributing to high workload for CHWs and long work hours.

In all three areas, the backgrounds of many CHWs indicate that they have undergone psychologically impactful experiences in various ways. One CHW from Shan hinted awareness of his symptoms of possible psychological trauma, and a few CHWs from Bangladesh expressed their awareness of their mental health needs and sought activities to occupy their minds and forget their past.

## Community level

**Feeling needed by the community.** For CHWs from Kayin, feeling needed by their community appeared to be the main reason for staying in their job. All CHWs except one demonstrated consistent dedication, expressing no intention to leave their service.

*"With these small achievements, I cannot leave them yet. This is because I have no specific person to hand over my responsibilities to for the time being . . . Now is an important moment with a difficult situation, and I cannot leave them . . . I will retire when my breathing stops."*
**60-year-old female Senior Medic, Kayin**

Six CHWs from Shan believed they could contribute to the healthcare needs of the community by serving there. One admitted experiencing moments when he contemplated leaving his job when his treatment skills did not satisfy a community leader. However, later, community members requested their authority to continue assigning him to their area.

In Bangladesh, some CHWs stated feeling needed in their job or that they can contribute something to their community as one other reason for them to stay in their job. Two CHWs expressed they want to continue to fill the gaps of their community with their current work.

**External level.**   *Safety and Security*. In Shan, one CHW expressed that, due to safety concerns associated with her job, she had an interest in pursuing a different job in the future. Another CHW from Shan mentioned that his parents no longer wanted him to work as a CHW due to safety and security concerns.

A leader within the Shan organization mentioned that they conducted interviews with both CHW candidates and their parents as part of the recruitment process. This additional step is taken because some parents express concerns about safety and security, hindering CHWs from going to their assigned areas even after completing training.

Even though it was not expressed as a factor affecting their motivation to continue their jobs, CHWs from Kayin and Bangladesh stated that they have safety and security concerns at work. Due to active conflicts in Kayin and violent actors in Bangladesh, they need to remain vigilant.

*Impact of Donors and International Non-Governmental Organizations*. In Kayin, one CHW mentioned that their salaries and access to medicines and medical supplies relied heavily on funding support from donors and International Non-Governmental Organizations. Consequently, delays in the procurement of supplies adversely affected their work.

Leaders from both Kayin and Shan organizations highlighted substantial funding gaps, impacting various aspects such as incentives for CHWs, recruitments and training programs. Additionally, the direct support for medicines and medical supplies from donors and INGOs were often insufficient and, at times, significantly delayed.

## Discussion

Utilizing visual elicitation techniques, such as the lifeline method, has been shown to enhance data collection by uncovering insights that might not emerge through verbal communication alone. These techniques foster deep conversations and reflections, thereby improving the quality of the data collected [31]. Specifically, the lifeline method complements life story/life history interviews, yielding richer and more comprehensive data [34]. This method has been effectively applied with health workers and community health workers in various global contexts [32,35].

Our study reaffirms the value of the lifeline method in the context of community health workers in Myanmar. By facilitating meaningful interactions and in-depth discussions, this approach bridges the gap between participants and interviewers. It underscores the potential of lifeline tools in qualitative research across diverse healthcare settings, although their benefits may not always be immediately apparent in unfamiliar contexts.

Findings from this study showed that acquiring skills and knowledge is a prevalent reason for both the motivation to become and continue as a CHW in all three study areas. For CHWs

in Bangladesh, the focus was on job-related skills and health knowledge. Meanwhile, for those in Kayin and southern Shan areas, the emphasis was on healthcare skills and knowledge. This aligns with previous findings from reviews and studies on the motivations of CHWs. These findings indicated that CHWs were motivated by opportunities for personal growth and professional development [36–38].

Excluding the organization in Bangladesh, the remuneration provided to CHWs did not meet their basic needs. As a result, it was not a motivating factor for joining the organizations in the Kayin and Shan. However, it was identified as a crucial factor for retention across all three areas, a finding that aligns with existing literature where CHW turnover was significantly related to low, no, or irregular compensations [39,40]. The organizations from the Kayin and Shan are already aware of this, but they do not seem to have enough funding to make significant changes. Most of them appear to mitigate this by focusing on recruiting single individuals who can manage with minimal support to be CHWs. However, this leads to CHWs leaving after marriage. CHWs who rely on support from their families also feel uncomfortable after seeking long-term support, and they aspire to be self-sufficient. In addition, some need to leave when their families require additional income.

Feeling needed by their community emerged as another significant and common factor for CHWs' retention which is in line with the motivational factor of wanting to help their own community that was found in all three areas, and supported by findings from other studies [36,41,42].

Relationships with community members and leaders contributed significantly to this aspect. In order to meet the expectations and needs of these community members and leaders, it is essential for CHWs to possess adequate skills and receive sufficient support from their respective organizations or partner organizations. This support should encompass not only sufficient medical supplies and timely logistical assistance, but also referral support, comprehensive pre-service and in-service training, effective supervision, and a conducive working environment [36,41,43].

A few health workers from Shan areas are addressing gaps in medicine and referral fees by paying out of their own pockets, which adds an additional burden given their insufficient remunerations. This was identified as a negative factor for CHW retention in a previous study [39].

Supervision and management have been identified as positive motivational factors in both Bangladesh and the Kayin study areas. However, a single yet significant finding from the Shan study indicates that supervision and management were perceived negatively, serving as a demotivating factor for continuing their work. This indicates that despite a lack of proper remuneration, a good working environment coupled with effective supervision and management can enhance CHW retention which is aligns with the findings from former studies [36,38].

While the above-mentioned factors for CHW motivation and retention are consistent with previous studies, this study reveals that these factors are intensified by the additional challenges faced by CHWs in conflict-affected areas. Safety and security concerns due to ongoing conflicts were prevalent across all study areas, adding stress for CHWs and, in some instances, prompting them to contemplate leaving their positions.

An emerging finding from this study is the potential need for mental health support for CHWs across all three study areas. Despite the well-documented impact of conflict on community-level mental health, few studies have examined its effect on CHWs [42,44]. Past unhealed traumas can also be exacerbated by insecure workplace situations. Moreover, inadequate compensation, or the absence thereof, along with a lack of future prospects in their jobs, are factors that can increase their stress. Working in conflict-affected regions with limited choices, facing dangerous situations, and operating in a challenging environment can further contribute to the mental health challenges of CHWs. The experiences shared by participants

across all sites in this study underscore a significant need for mental health support for CHWs, warranting further investigation.

CHWs from the study areas in Bangladesh have superior access to donor funding for their overall program, which enables them to meet the basic needs and expectations of CHWs more effectively. In contrast, in the study areas in Myanmar, CHWs has restricted access to funding and support for their comprehensive CHW programs. Moreover, the ongoing conflicts in Myanmar present substantial obstacles to the otherwise ideal strategy of integrating CHWs into the government health system for sustainability.

This study also discovered that organizations operating in these conflict-affected and fragile contexts face additional funding challenges, leading to shortfalls and affecting CHW retention. While these organizations recognized this as a management issue, they reportedly lack the necessary support, both technical and financial, to develop comprehensive CHW programs, as opposed to initiatives focused on specific training or projects.

The findings from this study underscore two primary needs: funding support for organizations and organizational support for CHWs. Particularly, organizations like the EHOs involved in this study need assistance to formulate long-term visions for their CHW programs, which can help reduce turnover and safeguard investments. This, however, calls for collaboration and commitment from both donors and organizations. Historically, CHW programs from conflict-affected areas were perceived as temporary. Yet, the period of stability in Myanmar from 2012 to 2020 demonstrated the persistent need for CHWs from these areas, whose services are irreplaceable [10]. Supporting evidence from other countries reveals a continued demand for CHWs in post-conflict settings due to a deficient healthcare workforce [39]. For enduring sustainability, the choice to invest in comprehensive CHW programs is vital for donors and international organizations. Additionally, further research should explore encompassing wider geographical areas with larger samples of CHWs in conflict-affected areas of Myanmar.

Conflict is a multifaceted social determinant of health that significantly impacts vulnerable populations in conflict-affected regions. Conflict disrupts health systems, infrastructure, and access to care. War zones, refugee camps, and areas affected by violence experience reduced access to essential services, exacerbating health disparities. The destruction of healthcare facilities, displacement, and insecurity further compound the challenges faced by affected communities [45,46].

Access to healthcare is pivotal for reducing health disparities. However, conflict disrupts the training, recruitment, and retention of healthcare professionals, posing significant challenges to effective health systems. In conflict-affected areas, CHWs play a crucial role in bridging gaps and providing essential care. To achieve Sustainable Development Goal 3 (SDG 3), which aims to ensure healthy lives and promote well-being for all at all ages [47], addressing conflict-related health challenges becomes paramount. Therefore, motivation and retention of CHWs are core components in achieving this goal in those conflict-affected areas.

## Limitations of the study

The funding and staffing of the research team by Community Partners International (CPI) could have potentially influenced participant responses due to power dynamics. However, as most participants were field-level operators in conflict-affected regions of Myanmar and marginalized communities in Bangladesh, they appeared either unaware or indifferent to CPI's relationship with their respective organizations. The use of the lifeline visual tool also seemed to mitigate the power imbalance.

A significant limitation of this study is the variation in roles and contexts among the three groups of CHWs compared. One group is restricted to health education and referral services and does not engage in patient treatment, unlike the other two groups. Additionally, while all

groups experience varying degrees of insecurity, the first group operates in areas with general violence and arsons but not active conflict zones, whereas the other two groups are more directly affected by conflict. These differences in responsibilities and operational contexts may affect the comparability of their experiences and perceptions, potentially influencing the study's findings and limiting the generalizability of the results.

Due to safety and security concerns, we were unable to collect data to the point of full saturation. Data collection trips within Kayin State required an approval process for each trip, while trips within southern Shan State incurred high expenses due to the need for hired vehicles. Additionally, our research was conducted with a small selection of organizations, which may limit its generalizability. Nonetheless, it is important to note that these organizations share similar structures and operations with many others in conflict-affected areas of Myanmar. The major themes identified in our study align with existing literature on the motivations and retention of CHWs.

Another limitation of this study was the use of single-coding due to resource constraints. Although the data was coded and recoded three times by the same researcher to enhance reliability, this approach could have influenced the interpretation of the findings.

Despite these limitations, the challenges and experiences uncovered in our research shed light on common issues faced by CHWs in conflict-affected regions globally. These insights can be valuable for implementing or improving CHW programs and understanding areas that require further investigation.

## Conclusions

This study highlights the unique challenges faced by CHWs in conflict-affected areas of Myanmar. It underscores how the operational context significantly impacts their motivation and retention factors. To address these issues, we propose that donors providing support for CHWs in conflict zones should offer more comprehensive assistance to EHOs. Additionally, programs should prioritize CHW mental health by conducting ongoing assessments of their needs and tailoring interventions accordingly. Engagement of stakeholders, including not only high-level leadership and donors but also CHWs themselves, is essential for answering the question of how best to sustain CHW programs and meet the needs of CHWs.

## Supporting information

**S1 File. Key informant interview guide: Question guide used for key informant interviews.**
(DOCX)

**S2 File. Life story interview guide: Question guide used for life story interviews.**
(PDF)

**S3 File. Selected abstract: Abstract selected for oral presentation in the 8th Global Symposium on Health Systems Research.**
(PDF)

**S4 File. Inclusivity in global research questionnaire: The inclusivity in global research checklist.**
(PDF)

## Acknowledgments

We appreciate Ler Htoo, Nang Hom Awan, Rumana Akter and partner organizations for their essential contributions to this research.

## Author Contributions

**Conceptualization:** Nyo Yamonn, Catherine Lee, Tom W. J. Y. Traill.

**Data curation:** Nyo Yamonn.

**Formal analysis:** Nyo Yamonn.

**Funding acquisition:** Tom W. J. Y. Traill.

**Investigation:** Nyo Yamonn.

**Methodology:** Nyo Yamonn.

**Resources:** Nyo Yamonn.

**Supervision:** Catherine Lee, Tom W. J. Y. Traill.

**Writing – original draft:** Nyo Yamonn.

**Writing – review & editing:** Catherine Lee, Tom W. J. Y. Traill.

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
