## [Decision Letter · Decision Letter 0]

28 Aug 2024

PGPH-D-24-01466

Insights and Inspirations: A Qualitative Exploration of Community Health Workers’ Motivations in Myanmar and Bangladesh

Dear Dr. Yamonn,

Thank you for submitting your manuscript to PLOS Global Public Health. After careful consideration, we feel that it has merit but does not fully meet PLOS Global Public Health’s publication criteria as it currently stands. Therefore, we invite you to submit a revised version of the manuscript that addresses the points raised during the review process.

There are some important review comments by the editor as well as a reviewer. Kindly go through them carefully and revise your manuscript. 

We look forward to receiving your revised manuscript.

Kind regards,

Vijayaprasad Gopichandran

Academic Editor

Journal Requirements:

2. Your current Financial Disclosure states, “The study is funded by Community Partners International's Core Funds.” However you do not provide a Funding information on the online submission form. Please indicate by return email the full and correct funding information for your study and confirm the order in which funding contributions should appear. Please be sure to indicate whether the funders played any role in the study design, data collection and analysis, decision to publish, or preparation of the manuscript.

3. Please provide separate figure files in .tif or .eps format.

4. In the online submission form, you indicated that "The datasets used and analyzed during the current study are available from the corresponding author on reasonable request. The data cannot be shared publicly due to data sensitivity.". 

a. In a public repository, 

b. Within the manuscript itself, or 

c. Uploaded as supplementary information.

5. Image.1.pdf: Please confirm (a) that you are the photographer; or (b) provide written permission from the photographer to publish the photo(s) under our CC-BY 4.0 license.

Additional Editor Comments (if provided):

The authors report a qualitative study that explores motivations and retention of CHWs in conflict prone areas in Myanmar and the Rohingya camps in Bangladesh. This is a crucial area of inquiry because health in conflict prone areas, politically unstable areas and refugee camps can be precarious and health workers play a major role in ensuring basic health in these areas. Their wellbeing, support systems and coping strategies are not studied adequately. Findings of such a study can help understand the situation of CHWs in conflict zones and in refugee camps.

The authors have provided very good description of the study settings in Myanmar and the Cox Bazar Rohingya refugee camp of Bangladesh. The exact details of who did the lifeline interviews, who did the lifeline visualizations, how and where they were done must be provided in detail. Reflexivity of the researchers and interviewers must also be provided to understand the process of the qualitative inquiry in detail.

Some more details of how the study participants were selected must be provided. How were these participant identified? Who identified them?

There is a good description about the study team in terms of the training provided to them. It would be useful to describe characteristics of the study team, their expertise, their level of experience and personal traits that might have an influence on the data collection, analysis and interpretation.

Was an interview checklist used? What probes were used to explore the lifelines? What materials were provided to sketch the lifeline? Did all participants draw their lifelines? Did any of them request support for drawing?

One of the researchers has coded and re-coded the data totally 3 times. The authors should provide details on which of them did the analysis? What are their characteristics? If they found a need to modify their code during subsequent coding sessions, how did they reconcile the difference?

Are names of the CHWs original or have they been changed to protect confidentiality?

Reviewers' comments:

Reviewer's Responses to Questions

**Comments to the Author**

1. Does this manuscript meet PLOS Global Public Health’s publication criteria? Is the manuscript technically sound, and do the data support the conclusions? The manuscript must describe methodologically and ethically rigorous research with conclusions that are appropriately drawn based on the data presented.

Reviewer #1: Yes

2. Has the statistical analysis been performed appropriately and rigorously?

Reviewer #1: Yes

3. Have the authors made all data underlying the findings in their manuscript fully available (please refer to the Data Availability Statement at the start of the manuscript PDF file)?

Reviewer #1: Yes

4. Is the manuscript presented in an intelligible fashion and written in standard English?

Reviewer #1: Yes

5. Review Comments to the Author

Reviewer #1: You have presented a well-structured protocol. I have a few suggestions:

1. Include more specific examples or case studies where similar methodologies have been successfully applied to enhance the justification of the approach. It will strengthen validation.

2. Address any potential limitations regarding the data collection process in conflict-affected regions, such as logistical challenges or data quality issues. Many can learn from this.

6. PLOS authors have the option to publish the peer review history of their article (what does this mean?). If published, this will include your full peer review and any attached files.

**Do you want your identity to be public for this peer review?** For information about this choice, including consent withdrawal, please see our Privacy Policy.

Reviewer #1: No

---

## [Editor Report · Decision Letter 1]

18 Sep 2024

Insights and Inspirations: A Qualitative Exploration of Community Health Workers’ Motivations in Myanmar and Bangladesh

PGPH-D-24-01466R1

Dear Dr. Yamonn,

We are pleased to inform you that your manuscript 'Insights and Inspirations: A Qualitative Exploration of Community Health Workers’ Motivations in Myanmar and Bangladesh' has been provisionally accepted for publication in PLOS Global Public Health.

Best regards,

Vijayaprasad Gopichandran

Academic Editor